# Service-based Analysis and Abstraction
# for Content Moderation of Digital Images

Moritz Hilscher*
Hasso Plattner Institute
Faculty of Digital Engineering
University of Potsdam, Germany

Hendrik Tjabben†
Hasso Plattner Institute
Faculty of Digital Engineering
University of Potsdam, Germany

Hendrik Rätz‡
Hasso Plattner Institute
Faculty of Digital Engineering
University of Potsdam, Germany

Amir Semmo§
Digital Masterpieces GmbH
Potsdam, Germany

Lonni Besançon¶
Faculty of Information Technology,
Monash University, Melbourne, Australia

Jürgen Döllner‖
Hasso Plattner Institute
Faculty of Digital Engineering
University of Potsdam, Germany

Matthias Trapp**
Hasso Plattner Institute
Faculty of Digital Engineering
University of Potsdam, Germany

## ABSTRACT

This paper presents a service-based approach towards content moderation of digital visual media while browsing web pages. It enables the automatic analysis and classification of possibly offensive content, such as images of violence, nudity, or surgery, and applies common image abstraction techniques at different levels of abstraction to these to lower their affective impact. The system is implemented using a microservice architecture that is accessible via a browser extension, which can be installed in most modern web browsers. It can be used to facilitate content moderation of digital visual media such as digital images or to enable parental control for child protection.

**Index Terms:** Computer systems organization—Client-server architectures; Computing methodologies—Image processing; Information systems—Content analysis and feature selection; Information systems—Browsers; Human-centered computing—Web-based interaction; Human-centered computing—Graphical user interfaces;

## 1 INTRODUCTION

This work's main objective is to support and facilitate human-driven moderation of digital visual media such as digital images, which are a dominant category in the domain of user-generated content, i.e., content that is acquired and uploaded to content providers. Content moderation usually requires humans who view and decide if content is considered to be appropriate or not, e.g., regarding violence, racism, nudity, privacy etc.; these aspects may have serious impacts in various directions, including legal and psychological issues.

To cope with negative impacts on the viewers' psychology and to alleviate legal issues, we propose a combination of content analysis and classification together with suitable image abstraction techniques that first detect inappropriate content and, subsequently, disguise and obfuscate content depictions or specific portions (Fig. 1).

---

*e-mail: moritz.hilscher@student.hpi.de
†e-mail: hendrik.tjabben@student.hpi.de
‡e-mail: hendrik.raetz@student.hpi.de
§e-mail: amir.semmo@digitalmasterpieces.com
¶e-mail: lonni.besancon@gmail.com
‖e-mail: juergen.doellner@hpi.de
**e-mail: matthias.trapp@hpi.de

### 1.1 Problem Statement and Objectives

From a technical perspective, the implementation of such approach needs to be independent of operation systems and processing hardware. Thus, we decide to use a service-based approach to detect and classify visual media content and to perform respective abstraction techniques depending on the detection results. Deep-learning approaches will be used that allow for defining what "offensive" means. Such functionality can be integrated into web browsers using a browser-extension based on a World Wide Web Consortium (W3C) draft standard. This way, the content moderation functionality can be applied and integrated into professional IT solutions or can be used by means of end-user apps.

**Facilitate Content Moderation (Objective-1):** Today, content moderation becomes more and more crucial for digital content providers (e.g., Facebook or YouTube) to fulfill their responsibilities and to implement an ethical content handling [9]. Moderation comprises manual examination for detection and classification of critical or inappropriate content. For some types of visual media content, the detection and classification can already be performed semi-automatically using machine-learning approaches. However, automated moderation being often limited (see e.g., [15]), the final moderation decisions are often made by humans, required to consume the unfiltered content.

Recent studies indicate that workers concerned with these tasks are often subject to severe mental health damage due to traumatic experiences or monotonic duties [12, 13, 31]. Interestingly, non-photorealistic rendering of these stimuli has shown promising results in reducing their negative impacts [1–3, 14, 39]. Further assessments of the effectiveness of such image processing techniques fall beyond the scope of this manuscript, which aims at providing a service-based implementation of what past literature has identified as helpful in mitigating the impact of shocking media. Therefore, these negative consequences could be mitigated by reducing the affective responses that arise from consuming the unfiltered content by using a combination of automatic analysis and abstraction techniques as follow (Fig. 2): (*i*) visual media content is analyzed, e.g., to detect, classify, and possibly perform a semantic segmentation, (*ii*) abstraction techniques are used to partially or completely disguise possibly offensive content prior to (*iii*), the interactive visual examination.

**Service-oriented Architectures (Objective-2):** To implement an approach for Objective-1 and to make it available to a wide range of applications and users, we set out to provide a prototypical micro-app (e.g., a browser extension) based on a microservice infrastructure. For this, two separate microservices, for analysis and abstraction respectively, are orchestrated by a content moderator

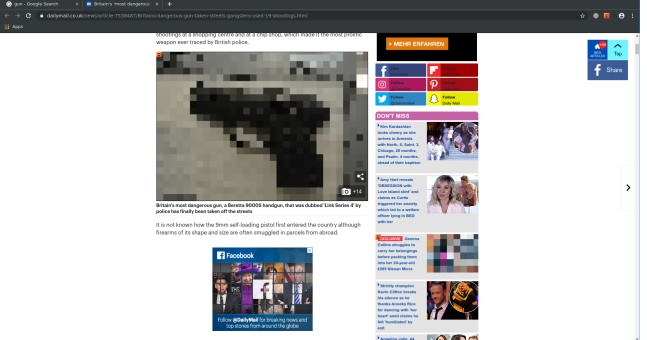

(a) Web-site with active browser extension for automatic content moderation.

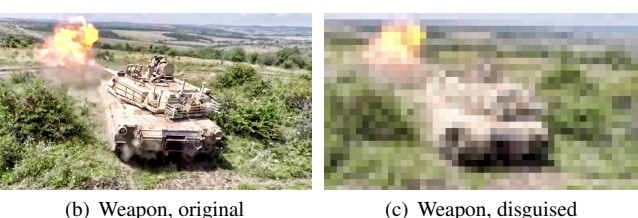

(b) Weapon, original      (c) Weapon, disguised

(d) Medical, original      (e) Medical, disguised

(f) Nudity, original      (g) Nudity, disguised

Figure 1: Application example for the combination of service-based analysis and image abstraction used for content moderation functionality provided by a browser extension (a). It enables the classification of digital input images (left) displayed on web sites according to different categories (rows) and their respective disguising using adjustable image abstraction techniques (right), such as pixelation (c), cartoon stylization (e), or blur (g)

.

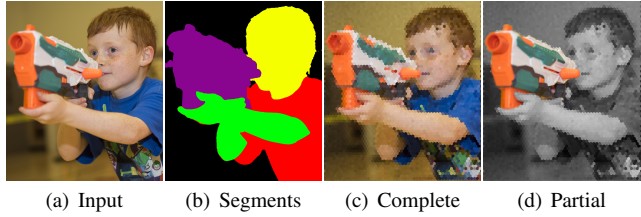

(a) Input    (b) Segments    (c) Complete    (d) Partial

Figure 2: Example of using image content analysis in combination with image abstraction techniques to disguise possibly inappropriate content for subsequent manual moderation: (a) input image, (b) results of image segmentation, (c) completely abstracted image, (d) partial abstraction techniques applied to the child only.

service. This enables a use-case specific exchange, replacement, or extension of specific functionality or complete services without risking the overall functionality. By using a Hyper Text Markup Language (HTML) User Interface (UI) integrated into a W3C browser extension, the abstracted content can be interpolated/blended with the original one interactively.

In current state-of-the-art systems, analysis and abstraction of images and videos are mostly performed using on-device computation. Thus, these systems' processing capabilities are limited by the device's hardware (Central Processing Unit (CPU) and Graphics Processing Unit (GPU)) and software (Operating System (OS)). Being subject to high heterogeneity (device ecosystem), this has major drawbacks concerning the applicability, maintainability, and provisioning of content-moderation applications, in particular: a software development process and integration into 3rd-party applications is aggravated by: (*i*) different operating systems (e.g., Windows, Linux, macOS, iOS, Android), (*ii*) heterogeneous hardware configurations of varying efficiency and Application Programming Interfaces (APIs) (e.g., OpenGL, Vulkan, Metal, DirectX), as well as (*iii*) display sizes and formats. Further, on-device processing does often not scale with respect to the increasing input complexity (e.g., number of images, increasing spatial resolution of camera sensors), which especially poses problems for mobile devices (e.g., battery drain or overheating).

## 1.2 Approach and Contributions

Concerning the aforementioned drawbacks of on-device processing, the proposed combination of standardized technology for micro-apps together with service-oriented architectures and infrastructures offers a variety of advantages, in particular: (*i*) implementation and testing of specific analysis and abstraction techniques are required to be performed only once (controlling the systems software and hardware due to virtualization), (*ii*) functionality is offered to a wide range of web-based applications using standardized protocols, which can be easily integrated into 3rd-party applications and extended rapidly. This way, (*iii*) the proposed service-based approach can automatically scale service-instances with respect to input data complexity and computing power required. Thus software up-to-dateness and exchangeability can be easily achieved. Further, the software development process of web-based thin-clients is less complex compared to rich-clients. Together with the upcoming 5G telecommunication standard featuring (among others) high data rates, reduced latency, and energy saving, the presented approach seems feasible to be applied in stationary as well as mobile contexts. Finally, intellectual property of the service providers can be effectively protected by not shipping respective software components to customers, and thus, impede the possibility of reverse engineering.

## 2 BACKGROUND AND RELATED WORK

### 2.1 Visual Content Analysis using Neural Networks

Image analysis can be performed according to different tasks. Image classification such as the ResNet Convolutional Neural Network (CNN) architecture can be performed [11] to determine how likely an image belongs to one or more specific categories. With object recognition, the goal is to identify objects displayed on images with it respective bounding boxes. For object recognition, CNN architectures such as is YOLOv3 [25] and Single Shot Detector (SSD) [19] can be applied. Another task is image segmentation, where objects and regions of certain semantics are identified on a per-pixel base. This can be performed with the approach R-CNN of Girshick *et al.* [10]. Different kinds of CNNs need to be trained with datasets consisting of images labeled with categories, objects, or image regions. There are public datasets available to train and to benchmark the performance of different CNN architectures. Popular examples are the Pascal VOC [6] and ADE20K [40] datasets. They

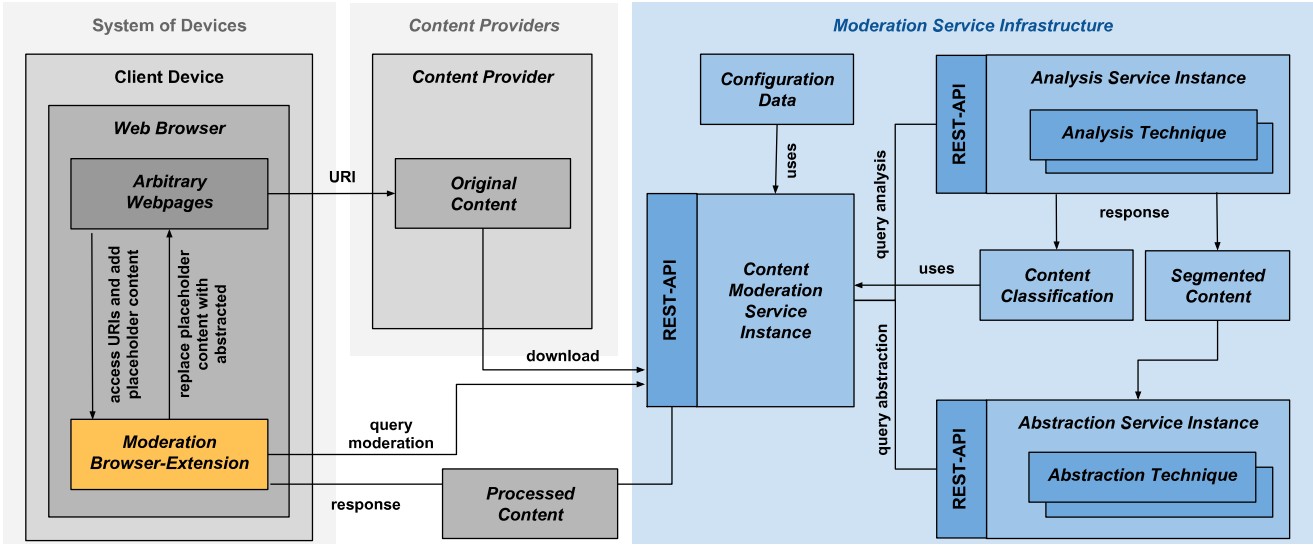

Figure 3: Conceptual overview of the microservice architecture of the presented approach. The individual service components (blue) are communicating via RESTful APIs and are used by a browser extension (orange) that integrates into standard web technologies (gray).

contain labeled image data for image classification, object recognition, and even semantic segmentation. For the task of content moderation, the objects and categories are quite general; mostly everyday objects are contained. Another dataset is the Google Open Image dataset [16]. It contains about 9 million images labeled with object information. The object categories are hierarchical organized and span over 6000 categories. Another approach, which spans even more different object categories, is YOLO9000 [24]. YOLO9000 is a variation of the YOLOv3 [25] architecture that was trained on a dataset with more than 9000 different objects. However, this dataset is not publicly available.

Further, there are approaches for image analysis that are more directed towards content moderation. These are mostly presented in the form of RESTful APIs allowing for sending images and receive analysis results. The Google Cloud Vision API [28] assigns scores to images depending on how likely they represent the categories *adult*, *violence*, *medical* and *spoof*. The API of Sightengine [30] analyzes images for the occurrence of categories such as *nudity*, *weapon*, *alcohol*, *drugs*, *scams* and other offensive content. Valossa [20] reviews cloud-based vendors supporting the classification of unsafe content and describes the difficulties of defining what inappropriate content actually is. They concluded that content analysis models must be able to understand the context of objects and depicted situations in order to decide whether images contain unsafe content. They offer an evaluation dataset with images in 16 different content categories and benchmark it on several online RESTful APIs. However, these approaches do not offer any public datasets or specify which machine learning models they use exactly. In contrast, Yahoo [21] offers a trained CNN model that can be used free of charge. The Yahoo Open Not Safe For Work (NSFW) model is basically a ResNet [11] that was fine-tuned on a dataset of NSFW images depicting nudity and other offensive content. For a given image, it determines a score how likely it contains unsafe content.

## 2.2 Service-based Image Processing

Several software architectural patterns are feasible to implement service-based image processing. However, one prominent style of building a web-based processing system for any data is the service-oriented architecture [33]. It enables server developers to set up various processing endpoints, each providing a specific functionality and covering a different use case. These endpoints are accessible as a single entity to the client, i.e., the implementation is hidden for the requesting clients, but can be implemented through an arbitrary number of self-contained services.

Since web services are usually designed to maximize their reusability, their functionality should be simple and atomic. Therefore, the composition of services is critical for fulfilling more complex use cases [17]. The two most prominent patterns for implementing such composition are choreography and orchestration. The choreography pattern describes decentralized collaboration directly between modules without a central component. The orchestration pattern describes collaboration through a central module, which requests the different web services and passes the intermediate results between them.

In the field of image analysis, Wursch *et al.* [38] present a web-based tool that enables users to perform various image analysis methods, such as text-line extraction, binarization, and layout analysis. It is implemented using a number of Representational State Transfer (REST) web services. Application examples include multiple web-based applications for different use cases. Further, the viability of implementing image-processing web services using REST has been demonstrated by Winkler *et al.* [36], including the ease of combination of endpoints. Another example for service-based image processing is Leadtools (https://www.leadtools.com), which provides a fixed set of approx. 200 image-processing functions with a fixed configuration set via a web API.

In this work, a similar approach using REST is chosen, however, with a different focus in terms of granularity of services. The advantages of using microservices are (*i*) increased scalability of the components, (*ii*) easy deployment and maintainability as well as, (*iii*) the possibility to introduce various technologies into one system [34]. For our work, we are extending a microservice platform for cloud-based visual analysis and processing that was first presented by Richter *et al.* [27]. In addition thereto and based on that, Wegen *et al.* [35] present an approach for performing service-based image processing using software rendering to balance cost-performance relation.

In the field of geodata, the Open Geospatial Consortium (OGC) set standards for a complete server-client ecosystem. As part of this specification, different web services for geodata are introduced [22]. Each web service is defined through specific input and output data and the ability to self-describe its functionality. In contrast, in the domain of general image-processing there is no such standardization yet. However, it is possible to transfer concepts from the OGC standard, such as unified data models. These data models are implemented using a platform-independent effect format. In the future, it is possible to transfer even more concepts set by the OGC to the general image-processing domain, such as the standardized self-description of services.

## 3 METHOD

With respect to Objective-2, we choose to implement our approach using microservices, which are described as follows.

### 3.1 System Overview

Fig. 3 shows a conceptual overview of the components as well as their data and control flow. It comprises of the following components that communicate via RESTful APIs:

**Moderation Browser-Extension:** A client-device running a web browser that (*i*) hosts the moderation browser-extension and (*ii*) an arbitrary website with visual media content. The website's visual media content is hosted by a content provider and referenced by an Unified Resource Locator (URL). The browser extension accesses the URLs via content-script and uses it to query the RESTful API of the CMS asynchronously.

**Content Moderation Service (CMS):** The CMS orchestrates the interplay between instances of the analysis and abstraction services, which encapsulate respective techniques. Upon request, it downloads the image from the given URL and forwards its content to an analysis service instance by querying the analysis RESTful API. Depending on the analysis response, it uses the configuration data to map analysis results to specific parameter values that are used to query the image abstraction service. The response is subsequently forwarded to the browser extension that replaces a placeholder content with the abstracted content.

**Content Analysis Service (CAS):** The CAS identifies if an image contains offensive content and has to be filtered using image abstraction techniques. It receives image data from the CMS and performs image analysis with different image recognition models as well as multiple image classification and object recognition CNNs. It then returns results of the different analysis models in a unified and structured way.

**Image Abstraction Service (IAS):** The IAS provides an interface for applying various image abstraction techniques (e.g., blur, pixelation, or more specific operations such as cartoon stylization, etc.) with presets of different strength to images that are identified by the CAS to possibly contain offensive content.

### 3.2 Browser-Extension for Moderation Client

The browser extension traverses the Document Object Model (DOM) tree and utilizes a MutationObserver object to detect changes in the respective image and picture tags; a DOM-MutationObserver is provided by the corresponding web browser and is intended to watch for changes being made to the DOM tree. As soon as an image is detected, it is initially blurred using Cascading Style Sheets (CSS) image filters. This prevents users to see possible disturbing image content while the CMS's RESTful API is queried and the image's URL is transmitted to it. The response received by the CMS contains information on whether or not an image contains disturbing content and therefore needs to be disguised. In the case that an image is categorized as offensive, the response also includes a processed version of the image.

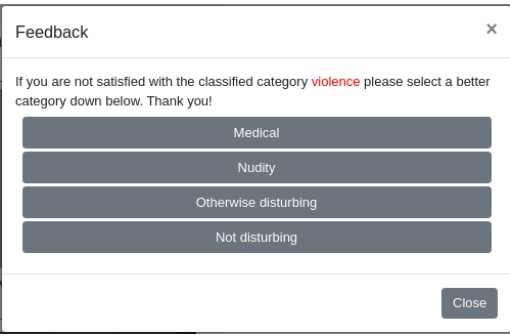

Figure 5: A feedback pop-up enables the correction of misclassifications and for feeding this information back to the server.

Subsequent to receiving the response, the local CSS-filter blur is removed. If the processed image does contain suggestive content, the original image gets replaced – otherwise, the original image is displayed. Finally, an overlay is added for every image (Fig. 4) that provides buttons for (*i*) allowing users to report misclassified images and (*ii*) toggling between the original and the processed image. If a user does not agree with the determined content classification, they can propose a more suitable one. A modal (Fig. 5) will appear where the user can select if another category is more suitable, the image is disturbing in a different way, or it should not be filtered.

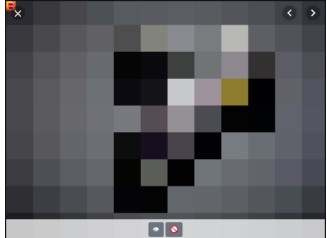

Figure 4: Filtered image with an overlay shown on mouse-over.

### 3.3 Content Moderation Service

The CMS moderates communication and interactions between the browser extension, the analysis service, and the image abstraction service. Clients use it to initiate the analysis process that consists of the following steps. First, the image is downloaded using the URL specified in the analysis request. Then, the image analyzer is queried to detect if the image contains offensive content. Subsequently, the CMS maps the image analysis results to an image abstraction technique and forwards it to the abstraction service for application. Finally, it notifies the client whether the image contains offensive content and, if it does, attaches the processed image to the response. If a user decides to send feedback via the feedback modal (e.g., the chosen scenario is unsuitable), a request to a feedback route is sent and the feedback is stored for further use.

### 3.4 Content Analysis & Abstraction Service

The CAS provides an interface for clients to analyze images for the presence of certain objects and categories. Clients send the image data and receive analysis results that are represented in the form of tags with score and meta data associated. Tags can comprise objects displayed in a image or categories that can be associated with an image. The score describes how likely an object or category is present. The meta data could include an Axis-aligned Bounding Box (AABB) that describes the estimated position of an object within the image.

The image analysis is performed with different machine learning models, in our case using CNNs. The output, specific to each model used, must be transformed to the unified analysis result format. This allows extending the analysis service with additional Machine Learning (ML) models. The results of the image analysis are used

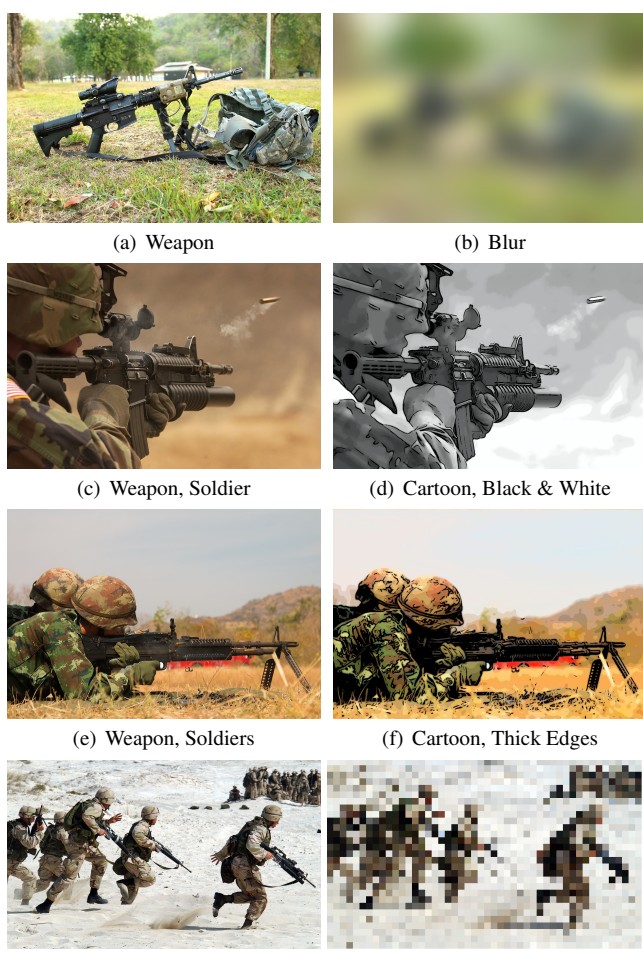

(a) Weapon

(b) Blur

(c) Weapon, Soldier

(d) Cartoon, Black & White

(e) Weapon, Soldiers

(f) Cartoon, Thick Edges

(g) Weapons, Soldiers

(h) Pixelation

Figure 6: Images with similar content (left) but different mappings (right) based on how likely they are rated as showing violent content.

by the content moderator to decide what kind of content an image is of and whether an image abstraction is applied.

If offensive content was detected on an image, it is disguised by applying an image abstraction techniques to it. The IAS provides an endpoint to apply specific techniques such as pixelation, or blur to an image. For every technique, different presets and parameters are provided to indicate different degrees and styles of image abstraction. To query the abstraction endpoint, the IAS requires the image's data as well as a mapped abstraction techniques and its preset (Sec. 4). With respect to this, the CMS perform such mapping by taking the analyzed scenario and score into account. In response, the processed image is returned by the IAS.

## 4   MAPPING OF ANALYSIS RESULTS

The analysis result for an image is a set of tags with scores. The tags describe objects that can be displayed or categories that can be associated with an image. The scores describe how likely these tags are actually present on an image. An image abstraction in the sense of content moderation, is processing a user generated content image with an image abstraction technique with a specific parameter preset. To process images with the goal of reducing explicit content, one has to define a mapping from analysis results to an image abstraction technique with a specific parameter preset.

In the proposed system each tag was manually associated with a scenario. A scenario is a type of content that should be moderated. For this system, the scenarios *nudity*, *violence*, and *medical* are used. Each scenario is specified by: (*i*) name, (*ii*) set of tag names and threshold, (*iii*) image abstraction technique, and (*iv*) three effect presets, sorted by degree of abstraction (low, medium, strong). A scenario is matched to analysis results if any of the scenario's tag names are contained in the received analysis' tags and the received score is equal or higher than the scenario tag score threshold. Because an analysis result could match multiple scenarios, the scenarios are prioritized and the matching scenario with the highest priority is chosen. If no scenario matches, then no image abstraction is required. Otherwise, the user-defined preset will be selected and used for the subsequent image abstraction step.

Fig. 6 shows four images with similar content but different mappings. All four images depict scenes with weapons and are categorized as showing violent content by the CAS. The used mappings are chosen according to the score that indicates how likely these images show violent content. Fig. 6(a) is disguised using a Gaussian blur preset with a large kernel size (Fig. 6(b)). Fig. 6(d) shows an applied cartoon filter using a black & white preset with thin edges to remove color information. Fig. 6(h) uses an cartoon filter with thick edges. Fig. 6(h) show the results of a pixelation abstraction technique for aggressive disguise. The effect that is mapped to an image is arbitrary and can be customized, but different abstraction techniques are more or less suitable for certain scenarios. In particular, this work uses a pixelation technique for images showing violent content, a cartoon filter for medical content based on the work of Winnemöller *et al.* [37] and as suggested by the studies of Besançon *et al.* [2, 3], and a Gaussian blur for images that depict nudity.

## 5   IMPLEMENTATION ASPECTS

In a service-based architecture similar to the one used in the content moderation scenario, it is necessary that messages are exchanged between the individual services. Therefore, each service provides a RESTful API and queries other services correspondingly. The browser extension is implemented using JavaScript (JS) and utilizes the used browser's API to access and alter a website's DOM tree to administer local storage and to react on changes made to the website. For sending requests to other services, the fetch API is applied. Filter options facilitate users to customize whether and to what extend suggestive images should be abstracted and all three scenarios can be customized individually. Users can choose among three levels of abstraction or can switch off single scenarios completely.

The CMS is implemented using Node.js and provides a RESTful API with two endpoints: one for requesting an image to be analyzed and categorized and one for sending user feedback. A request sent to the analyze endpoint needs to include the URL of the image that should be analyzed and options that represent the filter settings made by the user. A feedback request consists of three different pieces of information: the data concerning the assessed image, the category proposed by the image analyzer service, and the category included in the user's feedback. This data is stored and used as a training set for machine learning algorithms that can be used during image classification. The implementation of the IAS also relies on Node.js. It provides an endpoint that accepts requests that need to include the data of the image to be abstracted and an operation that should be applied to the image. A preset, related to the desired effect, can also be sent to this endpoint as an optional parameter. The CAS is implemented using Python and flask. It provides a RESTful API with an endpoint to start the analysis process.

For the basic functionality of the CMS, two different kinds of neural networks are used: (*i*) a Single Shot MultiBox model [19] and (*ii*) the Yahoo Open NSFW model [21]. Single Shot MultiBox is a CNN architecture that performs object recognition on images. For a given image it returns AABB with a scenario and a confidence

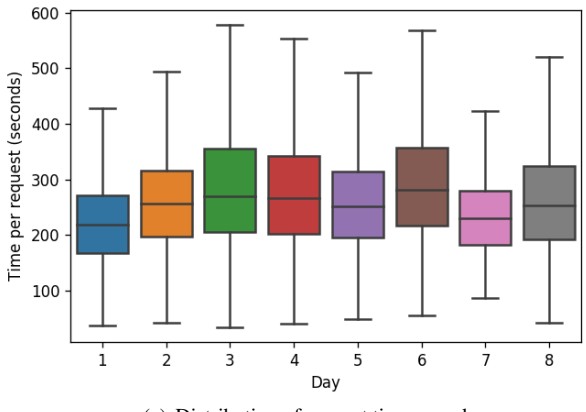

(a) Distribution of request times per day.

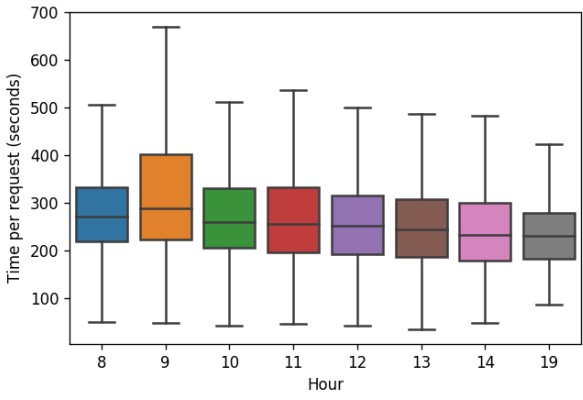

(b) Distribution of request times per hour.

Figure 7: Distribution of request times per day (a) and hour (b). Center line of boxes shows mean, boxes in total show quartile, whiskers show rest of the distribution without obvious outliers.

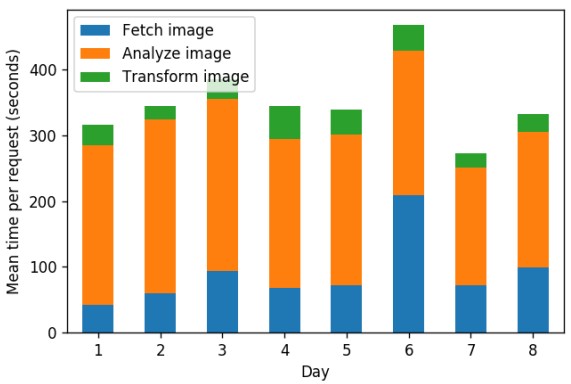

Figure 8: Mean request times according to different tasks.

score (0 to 1). The confidence score indicates to what extend a scenario is detected in the image. An existing implementation for PyTorch was used, as well as a classification model that has been initially trained on the Pascal VOC dataset [6], but with very general object classes such as aeroplane, car, cow, dog, or TV monitor. To this end, we also trained an SSD model on military-like classes of the Google Open Images Dataset [16] (such as rifle, tank, knife, missile) to be able to make predictions on somewhat realistic explicit content. The Yahoo Open NSFW model also returns for a given image an NSFW score (0 – safe, no nudity detected; to 1 – not safe, nudity detected) and an implementation and trained model for TensorFlow is available, but a proper threshold for this score is required. Such threshold might be different according to the use case of the system and must be chosen carefully.

## 6 RESULTS AND DISCUSSION

### 6.1 Performance Evaluation

We only focus here on system performance as the potential reduction of affective responses through image abstraction techniques has already been studied [2, 3, 39]. The system's performance was evaluated by timing different tasks involved in the process of content moderation and comparing these to assemble a metric of the requests and processed images (time stamp, image resolution, image transformation). Timed tasks measured on the CMS were (*i*) fetching image, (*ii*) performing image analysis, and (*iii*) performing image transformation by abstraction. The CMS, CAS, and IAS are hosted on a single dedicated GPU-server without a significant network overhead. Thus, the network request times between clients and the services are assumed to be independent of our system and are not considered. To evaluate the performance in a way that considers all required tasks equally, only requests that led to an image transformation (unsafe image was detected) were considered. The dedicated GPU-server is equipped with Xeon E5-2637 v4, 3.5 GHz processor (8 cores), 64 GB RAM, NVIDIA Quadro M6000 GPU with 24 GB VRAM.

Over a week of testing the extension, about 35 000 requests involving image transformation are logged in total. Fig. 7(a) and Fig. 7(b) show the distribution of total request times per days and hours. These are mostly independent of each day and hour with slight variances, which could be explained by a varying load of the server or different number of requests incoming in a shorter period of time. Fig. 8 shows the mean time required for each task per day. The image analysis requires $\approx 75\%$ of the mean request time, with some outliers on day 6 where fetching the images suddenly takes up as much time as analyzing the images on average. Fig. 9(a) shows the resolution of images over the time required for analysis.

We further tested whether images of high resolution impact analysis performance. The documentation of the used CNNs describe that image data is propagated through the networks at a constant resolution, i.e., a downsampling is required before propagation through the CNNs. Fig. 9(a) shows that images have similar analysis times independently of the resolution and the required downsampling step. A further question relates whether very small images (such as icons) cause a performance overhead if they appear on websites very often. Images that are smaller than $32 \times 32$ pixels are highlighted in red within Fig. 9(a). One can see that they need similar inference times compared to all other images. Further analysis of the statistics also indicate that they take up only $\approx 5\%$ in count of all images and less than one percent in total request time. A similar analysis was performed for the image transformation task. Relating to the image resolution vs. the image transformation (Fig. 9(b)) shows a linear dependency between image resolution (as number of total pixels) and transformation time. However, this does not impact the complete performance severely as image analysis is slower by a factor of about 10. Small images have been highlighted, again they take up $\approx 5\%$ in count and about $\approx 4\%$ in total time only.

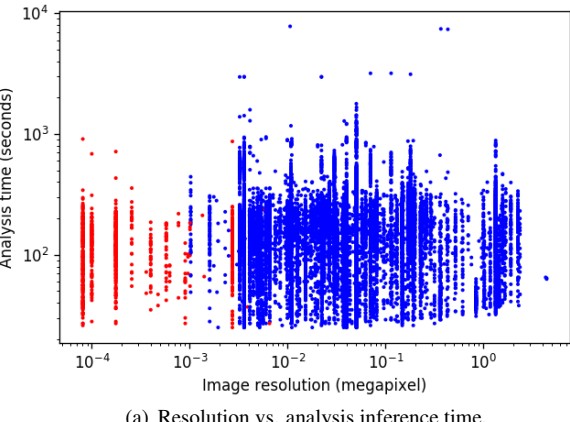
(a) Resolution vs. analysis inference time.

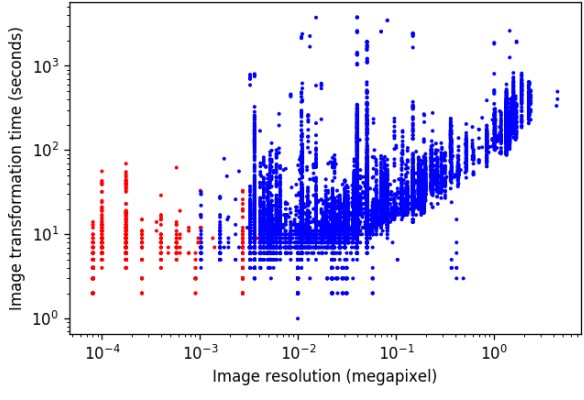
(b) Resolution vs. image transformation time.

Figure 9: Relating resolution of input images with analysis inference time (a) and image transformation time (b). Images smaller than $32 \times 32$ pixels are highlighted in red.

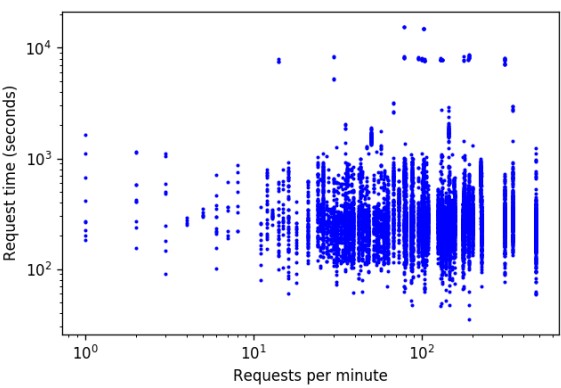

Figure 10: Relating requests per minute and the total request time.

Finally, we evaluated whether a high load (possibly caused by a high number of requests in a short time) causes higher total request times. For it, each request is plotted as a point with the number of requests in that minute and its required time (Fig. 10). This does not show any strong correlation between the number of requests per minute and the total request time though; perhaps, the load generated by the requests was not high enough yet.

## 6.2 Applications

The system described in this work offers advantages for the consumption, use, and eventually moderation of graphic content in different areas we now detail. First of all, it could assist for medical and surgical education. A primary use case would be to facilitate the education of nurses and medical students (not all destined to be surgeons) by reducing affection and aversion when looking at images showing blood and medical acts [2, 3]. Similarly, another use case is on the communication between surgeons and patients [2, 3]. Patients are usually informed and prepared before the planning of future surgeries and the explanations can be facilitated by the use of images. Yet, laypeople often find looking at images depicting surgery or blood extremely difficult [8, 29]. Communication between patients and doctors could therefore be improved with such automated image processing tools.

Furthermore, the system can be used to moderate internet forums and social networks. Nowadays, a lot of digital media, including images and videos, are shared through social media platforms such as Facebook or Instagram. While graphic content sometimes adheres to the Terms of Services (TOSs) of these platforms [7, 32], many graphic media are not accepted for ethical or legal reasons. The filtering between authorized and non-authorized content is performed by a combination of algorithms and people, or people alone (content moderators) depending on the platform. The software system presented in this paper could facilitate a content moderator's work and help to prevent mental issues that can be a consequence of looking at disturbing pictures all day [23]. The system could also alleviate the toll on volunteer moderators of platforms such as Wikipedia or Reddit [4, 15]. In a similar fashion, journalists and news editors might have to browse through hundreds of shocking content to illustrate their articles or better understand the case they are reporting on (e.g., war-zones, disasters, accidents). Our automated tool could also be useful in this specific context.

Finally, since exposure to graphic or pornographic content has been shown to be particularly detrimental to children [18], our tool could particularly be interesting on this regard. While blocking software could be used to limit access to nudity or pornography, such software tend to also limit access to useful information (e.g., online sex education) [26], are rarely maintained or even used [5], and unlikely to block access to content on some social media platforms. Blocking software finally rarely target all sorts of graphic content. With respect to this issue, we hope that our tool can help limit the impact of unwanted graphic content, rather than eliminate it completely along with potentially useful information.

## 6.3 Limitations

Applying specific filters to weaken affect when looking at medical images might work differently or not at all depending on the user and the specific image at hand. More generally, it is impossible to find a definition of offensive content that fits all users. What is perceived as "graphic" highly depends on the perception, age, views, cultural background, and personal history of the user, which results in many different potential use cases for each individual user. Even if a clear definition would be possible, modern computer vision approaches are not able to correctly recognize offensive content all of the time. Additionally, detecting objects that are known to be offensive is not enough. The context in which these objects are as well as the overall image composition could completely change the meaning.

We do not train accurate neural networks for real use-cases because there are no acceptable, publicly available datasets of offensive content. Even with a proper dataset, it is unlikely to train a perfect neural network, which classifies all images correctly and detects offensive content every time.

Regarding the browser extension, it is difficult to even detect all the images on websites since there are a number of different ways to integrate an image into a web page, e.g., through custom HTML elements or extensive JS usage. If images are loaded asynchronously via JS and many images change simultaneously, the extension is not able to react fast enough to all of the changes, resulting in "un-obfuscated" or "non-abstracted" images. Moreover, JS is executed in a single thread in all established web browsers, which increased the occurrence of time-related problems. Additionally, a very low bandwidth could be slowing down the processing of images. The impact would be comparably tiny because not much additional data is sent and the client still downloads each image only once.

## 7 CONCLUSIONS AND FUTURE WORK

This paper presents a service-based approach to facilitate consumption of digital graphic images online. To achieve this, an automatic analysis and classification of possibly offensive content is performed using services, and, based on the results, image abstraction techniques are applied with varying levels of abstraction. This functionality is accessed and configured via a browser extension that is supported by most modern web browsers. The presented content moderation approach has various applications such as reducing affective responses during medical education, allowing less distressing browsing of social media, or enabling safer browsing for child protection.

Regarding future work, the user experience of the extension could be increased as follows. Modern object recognition algorithms are not only able to detect certain objects in images but are also able to locate them. With respect to this, image abstraction techniques can be applied to segments of the image to maintain the context and only abstract the sensitive image regions. This might support the user to identify whether an image was classified correctly. As an alternative, the image abstraction techniques could be applied to the complete image, but the user could be given the option to interactively reveal the image partial using lens-based interaction metaphors.

### ACKNOWLEDGMENTS

This work was funded by the German Federal Ministry of Education and Research (BMBF) through grant 01IS18092 ("mdViPro") and 01IS19006 ("KI-LAB-ITSE").

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
