# OpenReview forum: "Service-based Analysis and Abstraction for Content Moderation of Digital Images"
_graphicsinterface.org/Graphics_Interface/2021/Conference/Second_Cycle — GI 2021_

### Official Review · Reviewer_tfaX · 2021-04-30
**Service-based Analysis and Abstraction for Content Moderation of Digital Images**

**Rating:** 3
**Confidence:** 2

**Review:**

This paper describes a service-based architecture that can classify images on the Web as inappropriate and apply filters and other image manipulations to reduce the “affective” impact of these images.  The implementation is validated through an experiment showing the speed at which the system can do its work.

This is a clearly written paper, but unfortunately, I do not see the contribution as being appropriate for an HCI audience.  While ways to deal with inappropriate content have the potential for relevance to the HCI community, no human-centric perspectives are directly addressed in submission, beyond the high-level motivation for the framework.    In my opinion, the current focus on implementation of existing techniques and speed-based validation, requires a different set of reviewers who can assess the strength, novelty, and validity of the combining the techniques in this manner.  The paper would also need to reconsider to related literature, to help motivate and position the intended HCI contributions.  The fact that most of the submission’s references are to papers in computer vision or graphics venues is another indicator of needing the better articulate and motivate the contributions.

For future submissions, I would also recommend reconsidering the images used for demonstration purposes. Including an image of a women in a bikini on the first page and further claiming that she is “naked” is likely not going to resonate with many reviewers concerned with EDI and the representation of women in our scientific papers

---

### Official Review · Reviewer_Kmg5 · 2021-04-30
**service-based digital image moderation**

**Rating:** 6
**Confidence:** 4

**Review:**

This is an interesting study which looks at service-based digital content moderation. Overall the study is well presented and is well structured, with a good background section and clearly developing a system which has much real world value and potential to improve online usage scenarios for a wide range of users and use cases. Although the idea appears novel and possibly may be effective, the evaluation of the proposed system only focuses on system performance and does not attempt to, for example, evaluate to a reasonable level the content moderation from a user perspective. I do not feel that  the value, effectiveness or contribution of the proposed system can be seen as being significant given the very constrained criteria that have been used to evaluate the proposed system. Overall, an interesting study, but with a rather constrained and one-dimensional evaluation which unfortunately leaves the reader with many questions and not many answers about the overall contribution.

---

### Official Review · Reviewer_QM9F · 2021-05-04
**The paper is good but requires minor modifications**

**Rating:** 7
**Confidence:** 3

**Review:**

The paper presents a new service-based approach to moderate digital images on web pages. The authors report on the system implementation and the system performance evaluation.
Overall, I found the paper to be very well-written, with a clear structure, clear sections separation and naming, helpful illustrations, helpful itemized structure of important points, etc. One comment that I had regarding the readability of the paper is that the structure and order of subsections in section 6 are a bit confusing. In particular, the paper could probably benefit from moving the discussion of applications (section 6.1) after the performance evaluation, closer to the overall discussion of the work and its limitations.
Among the strengths of the paper, I would also want to mention the clarity of the overview of related systems, approaches, and overall, the related research. Although I am not a specialist in this particular question, I found the overview to be clear, descriptive, convincing, and well built into the narrative. On a similar note, I think the paper does a good job at outlining the motivation for this research and at grounding it in previous work. However, I would like to mention that a lot of motivation points and application discussion aspects are built around the effects of images on viewers. Although authors mention that they do not focus on the assessment of the effects of the described system on the affective perceptions, I would recommend extending the discussion of previous findings to provide readers with more context and to balance the human-centered view, strongly articulated in other sections. I also found the discussion of application to be a bit disconnected from the overall story in the paper. Perhaps, slightly more elaborated motivation for the suggested applications could be helpful.
In general, I think the length of the paper is fitting the size of the contribution, and I believe that the GI community can benefit from this paper.

---

### Meta-Review · Area_Chair_nQUQ · 2021-05-04

**Recommendation:** Accept
**Confidence:** 1

**Metareview:**

The reviewers agree that the paper is well written.  Reviewers also felt that the problem is important, well-motivated and grounded in literature.

Reviewers also agree on the disadvantage of the paper, which is a weak evaluation and a lack of human-centric considerations in the latter half of the paper.

Given that two of the reviewers see enough value in the paper to recommend accept, I will go with the majority, particularly given that the other two reviewers are more confident in their assessments.  I like the suggestion to strength the discussion in a way that brings in a human-centric element again.  I see this as a reasonable way to address my concerns.

---

### Decision · Program_Chairs · 2021-05-08

Accept